# Development of Students' Learning to Learn Competence in Primary Science

## Alena Letina

Faculty of Teacher Education, University of Zagreb, 10 000 Zagreb, Croatia; alena.letina@ufzg.hr

**Abstract:** The aim of this study was to determine if there is a connection between the inquiry-based teaching of science in primary school and the development of the students' learning to learn competence. The research involved 333 fourth-grade students at primary schools in Croatia. The experimental research with parallel groups included a number of students who were exposed to inquiry-based teaching of science for three months, and a control group that was, at the same time, exposed to traditional instruction in the same curriculum content. The results of this research show that through inquiry-based teaching students developed a higher level of the learning to learn competence than by using traditional teaching methods. Therefore, it is recommended to use inquiry-based learning as often as possible, because by developing the students' learning to learn competence, students will be empowered for the process of lifelong learning.

**Keywords:** inquiry-based teaching; learning to learn competence; self-regulated learning; students' perception of learning

## 1. Introduction

In the contemporary educational system, a high place in the hierarchy of educational goals and key competencies needed by students to successfully cope with everyday life situations and problems is taken by the students' ability to independently manage and regulate their own learning process. Due to a central role it has in achieving the quality of learning and student performance in and out of school, self-regulated learning or the learning to learn competence has become one of the key constructs in education [1].

Besides understanding and learning skills, the learning to learn competence encompasses attitudes, values, and beliefs that enable a person to develop efficiency, flexibility, and self-organization in learning in a variety of contextual frameworks [2]. Based on these characteristics, it can be defined as a meta-competence because it has a significant impact on the acquisition and application of other competencies. The reason why the European education policy focuses on learning competence as one of the key competencies that every European citizen should develop stems from the accelerated global changes, prompting educational activities to prepare students for coping with these changes and train them for lifelong learning.

Learning to learn is a process which focuses on an individual's self-awareness as a student, which includes one's motivation to learn, one's learning goals, preferred learning strategies, and cooperation with other students. During life, especially during the intensive process of education, the individual, mostly unconsciously, develops awareness of himself/herself as a student and, on the basis of that awareness, shapes his/her learning strategies. The learning to learn competence implies awareness of the concept of learning and the process that takes place in its essence, as well as the ability to adapt that process if some limitations occur. The learning to learn competence involves entering into the deeper meaning of the structure of a particular material during the process of learning and can lead to critical awareness of the assumptions, rules, and social expectations that affect human

cognitive experience, as well as their way of thinking, feeling, and behaving during learning [2]. This competence relates to motivation for learning, learning goals, preferred ways of learning, learning strategies, and cooperation with others [3] and allows students to become more effective, flexible, and self-organized learners in a variety of contexts [4].

The learning to learn competence also includes the ability to organize and structure one's own learning, in individual or group contexts, as well as the ability to effectively manage time and information, problem solving, and adoption, application, and evaluation of new knowledge in different circumstances [5]. It includes awareness of the learning process and the need for learning, as well as the ability to overcome obstacles for more effective learning. It also involves the assimilation, application, and evaluation of new knowledge and the application of acquired knowledge and skills in different contextual frameworks. In a broader sense, it can significantly contribute to the personal and professional development of a person. It is clear from this definition that this competence encompasses both the cognitive and affective dimensions and indicates its transversal nature and its lifelong and continuous dimension. The basic knowledge, skills, and abilities involved in this competence are shown in Figure 1.

**Learning to learn competence**

It includes the availability and ability of a student to organize and regulate his/her own learning, either in individual or group contexts, and the ability to effectively manage learning time, solve problems, and assimilate, apply, and evaluate new knowledge, as well as the application of the acquired knowledge and skills in different contextual frameworks.

| **KNOWLEDGE** | **SKILLS** | **ATTITUDES** |
|---|---|---|
| Knowledge and understanding of different learning methods, strengths, and weaknesses of their own learning skills. | Ability to self-regulate learning, effective management of learning time, autonomy, discipline, perseverance, and information in the learning process. | Positive attitude towards learning and readiness for further development of learning to learn competence, motivation, and confidence in one's own success during learning. |
| Knowledge of educational opportunities and understanding how decisions during education lead to different professional careers. | Ability to concentrate in longer and shorter periods of learning. | Positive attitude towards learning as an activity that is important for the life of every person and the development of initiative for learning. |
| | Ability to think critically about the goal and purpose of learning. | Flexibility in the learning process. |

**Figure 1.** Basic knowledge, skills, and attitudes which are part of the learning to learn competence.

These components of the learning to learn competence indicate its great complexity. Under the influence of the rhythm of contemporary social changes, learning is no longer a one-time acquisition,

then repetition and maintenance of the already acquired knowledge but takes on the characteristics of an innovative activity that constantly creates something new [6].

Learning to learn is neither a set of skills nor a recipe that will enable the improvement of learning. It is a kind of philosophy that puts the student in the center of attention through several key factors: (a) the school, which provides the student with the opportunity to learn; (b) parents, who participate in the shaping of the learning process by encouraging children to find their own methods and strategies of learning; and (c) students themselves, who seek to develop into individuals who will practice lifelong learning throughout their lives.

Conceptually, learning can be divided into two broad categories: learning as the reproduction of knowledge and learning as the transformation of knowledge. The first category includes the understanding of learning as an accumulation of knowledge (increasing the amount of information), or as a process of memorizing knowledge and skills with the aim of using them later. The second category refers to the notion of learning as a process of discovery that enables understanding of the phenomena in nature and an activity that leads to conceptual (and personal) change. According to the above, in order to develop the learning to learn competence within a lifelong perspective, it is necessary to consider learning as a process which does not involve mere memorization and reproduction, but knowledge transformation [6].

Research on the learning to learn competence, its basic characteristics and key elements, and its development among students has become more frequent in the last decade. Thus, within the Finnish project "Life as Learning", the University of Helsinki organized a series of studies dealing with the study of this competence [7]. Thanks to the British project "Teaching and Learning Research Program", a number of questions related to this competence were also opened [6]. The University of Bristol launched a project known as the "Effective Lifelong Learning Inventory" (ELLI) [8], which aimed to define and examine the personal orientation of respondents towards lifelong learning. They used "learning power" as a new term that implies a complex mix of disposition, experience, social relationships, values, attitudes, and beliefs that influence a person's individual engagement in different learning opportunities [9].

There are other attempts to define this competence, such as the British Learning Campaign, which defines the learning to learn competence as a process of discovering the learning itself that enables students to learn more effectively [10]. The Centre for Research on Lifelong Learning (CRELL) as a part of the Institute for the Protection and Security of Citizens of the European Commission, in debates related to defining the learning to learn competence, started from the concept of metacognition. In this context, Bakračević [11] defines metacognition as an important component of the learning to learn competence, and Sorenson [12] emphasizes the perspective of metalearning as an essential feature of this competence, where metalearning refers to learning how to learn. Furthermore, Black et al. [13] defines the learning to learn competence as a combination of knowledge of cognition (knowledge of what a person knows and does not know) and self-regulatory mechanisms in learning (such as planning the learning process, checking outcomes resulting from the application of a particular learning strategy, assessment of these outcomes, and revisions of the strategy to improve the learning process).

When it comes to this competence, the term self-regulated learning is often mentioned. Self-regulated learning, as a type of competence, implies a multi-component, cyclical, self-initiated process that involves cognitive, metacognitive, and motivational systems; behavior; and adaptation of learning situations in order to achieve student goals [1]. Bakračević [11] emphasizes that self-regulation, along with metacognition, is an important part of the learning to learn competence, and Moreno [14] talks about certain elements of metalearning, such as planning and monitoring the learning process, that can be described as self-regulation. Self-regulation is considered to exceed metacognition because it includes affective, motivational, and behavioral monitoring and self-control processes [15].

Attempts to measure the learning to learn competence are found in some of the already mentioned projects such as the University of Helsinki project, the University of Amsterdam cross-curriculum test, and The Effective Lifelong Learning Inventory project (ELLI) of the University of Bristol. The University

of Helsinki project designed a test that sought to cover both the cognitive and affective dimensions of this competence and the sociodemographic characteristics of respondents and later included examinations of teachers' attitudes to the development of the learning to learn competence and teaching methods [16]. Using this instrument, numerous surveys were conducted in Finnish schools in the period from 1996 to 2006, which enabled a longitudinal study and comparison of the students' learning to learn competence and its development during a series of educational cycles.

The University of Amsterdam also became involved in measuring the basic features of this competence by designing a cross-curricular skill test (observation, selection, and editing of information; summarizing and reasoning; forming opinions; recognizing beliefs and values in opinions; distinguishing opinions and facts; collaborative learning; and expectations of the quality of their own work). The University of Bristol developed an instrument to examine the effectiveness of learning by measuring the "learning power" of each individual student. A key feature of this instrument is the examination of the seven factors which are a part of the learning to learn competence and the ability of teachers to apply it for diagnostic purposes to make their students aware of the learning process [9]. These factors are: growth orientation (perspective of learning as a lifelong process), meaning-making (making sense of new things in terms of previous experiences), critical curiosity (enjoying learning challenges), fragility and dependence (preference for less challenging situations in learning), creativity (playing with ideas and taking different perspectives), learning relationships (managing the balance between being sociable and being private in their learning), and strategic awareness (interest in different approaches to learning).

The theoretical starting point and framework of the European test for examining the learning to learn competence consist of the cognitive, metacognitive, and affective dimensions of this competence. The affective dimension consists of three sub-dimensions. The first sub-dimension consists of the motivation to learn, learning strategies, and orientation towards change; the second consists of academic self-confidence, whereas the third includes the environment in which learning takes place. The cognitive dimension also encompasses several sub-dimensions. These include identifying statements, applying rules, examining rules, and applying mental tools. Finally, the metacognitive dimension includes problem solving, metacognitive precision, and metacognitive self-confidence. The preliminary results of the research with the application of this instrument indicated the need for its further improvement.

One of the most well-known international tests of student knowledge and skills so far is the Program for International Students Assessment (PISA), which aims to determine "what students can do with their knowledge". The survey does not focus on any particular aspect of the curriculum but seeks to assess how well students can use knowledge in everyday life situations. In this context, PISA approaches the examination of certain aspects related to the learning to learn competence, and the tasks used show similarity to the framework for testing this competence [17].

Training students to apply different learning methods and techniques should begin from the first level of formal education. In the contemporary pedagogical literature, such an attitude is often accompanied by the use of the term metacognition [18]. This concept refers to a person who self-consciously explores his/her mental processes while simultaneously adapting and improving the effectiveness of their learning [19]. The notion of metacognition as a synonym for the learning to learn competence is applied by other authors [20] in addition to Flavell and Childe [18,19]. Under this term, they include understanding the learning process, students' awareness of their own learning characteristics, perceiving the conditions, and manipulating the context content in order to achieve a higher level of efficiency in learning.

For the development of this competence, students must first ask themselves why they are learning and how the things they are learning can serve them. A number of organizational preconditions should then be taken into account for the creation of basic learning conditions: preparation of the workplace and learning materials, review of the content, and planning of the required time. This is followed by planning the work on the task, applying the technique of reading and taking notes, separating important things from the irrelevant, applying memory techniques, and measuring learning outcomes.

These are the basic preconditions for forming metacognition of one's own learning. Each of these assumptions is accompanied by metacognitive insights that children gain in specific learning situations. Once these metacognitive insights are realized, it can be expected that there will be a transfer from one content to another or one situation to another. There are four typical elements for an effective transfer of metacognition about learning: (1) knowledge which is common to new and old tasks, (2) skills and abilities that can be used on a new task, (3) learning habits, and (4) students' characteristics that can generally help in the learning process such as persistence, competencies, and enjoyment in learning [21].

Children do not generally learn to view the components of their learning on a metacognitive level. This insight is most often prevented by their prejudices or implicit theories about learning. These are beliefs that prevent children from successfully working on the task because they believe that they are not capable of mastering the content or they believe that they can master this content easily.

The learning to learn competence is the essence of the educational process. It strongly influences the ability to manage a professional career [3] and its development is therefore very important for every student. It is sometimes equated with lifelong learning and requires the development of metacognitive skills [22]. The usefulness of the education system for students would certainly be greater if students, in addition to acquiring more and more complex specific knowledge, also developed the general skills necessary for an effective approach to the learning of various contents. Today, specific knowledge and skills are rapidly becoming obsolete and therefore need to be constantly renewed. Without a properly developed learning to learn competence, a person is exposed to an increased risk of social and economic exclusion [5].

Research in the area of higher education has shown that students often have inappropriate learning habits and prejudice towards learning [23] and that they lack the self-assessment and metacognitive skills required for self-identification of problems in their learning strategies [24].

The importance of the development of this competence was emphasized by Higgins [25], who claims that learning to learn should become a key feature for the future of education, as it would allow students to realize their potential.

The results of research on the presence of key learning to learn competencies and entrepreneurship among students in Croatian primary schools [4] showed a clear need for the improvement of the learning to learn competencies in primary school students. The effective application of learning strategies, the development of a positive motivational basis for learning, and the development of the habit of regular learning as a basis for these competencies need to be encouraged by systematic measures at the level of the entire education system. Inquiry-based teaching is one of the strategies by which the mentioned competence can be supported and developed. Its goal is not only to develop the students' research competence and the researcher's mindset but also to foster metacognitive competencies, which include experiences and processes connected with the control of students' own cognitive functions [26].

The low level of development of self-regulated learning among students is partly a reflection of the methods of learning and teaching in schools [4]. Therefore, it is necessary to use contemporary teaching methods and assessments of learning to encourage students' achievement.

One of the basic problems in learning how to learn is students' motivation for this activity [24]. Orientation to learning should have a deeper meaning and significance in the child's world of values. Like any intrinsically motivated activity, learning to learn should have a meaningful context. The child wants to learn and has a natural need to learn. McDougall [26] calls this need an instinct of curiosity. This instinct is accompanied by the emotion of wonder; Berlyne [27] calls it the instinct of curiosity or exploration. Analogously, Pinker [28] says that children are born with a "learning instinct". Children love to learn but learning should be like a game. Learning, as the joy of discovery, is usually accompanied by a sense of satisfaction. It is therefore justified to ask why this natural need of the child has become a hated and uncomfortable activity for children, because students very often do not express too much inclination towards school learning, especially not toward learning based on

a traditional teaching concept. The answer can be found in formalism, verbalism, the dominance of frontal teaching, and other characteristics of the traditional school.

However, despite all the known facts about the benefits of this approach, teachers tend to teach students rather than introduce them to a complex learning system. In this context, it is important to emphasize the need to build learning pedagogy, a pedagogy that will train students not only to remember facts and content but also to find, use, and store information; to separate the important from the irrelevant; to valorize what has been learned; and to ask questions about one's own cognition.

This study will show experimental research results, aiming to determine if there is a correlation between inquiry-based teaching of science and the development of students' learning to learn competence. The research objective was to investigate the effect of inquiry-based versus traditional (lecture-based) teaching on students' motivation to learn science, their perceptions and learning habits, their adaptation to different learning circumstances, and the frequency of using different learning strategies in order to increase its efficiency. Following previous findings on the benefits of inquiry-based teaching, the authors hypothesized that inquiry-based teaching of science in primary school would result in a rapid development of the students' learning to learn competence compared with traditional teaching.

## 2. Materials and Methods

### 2.1. Research Sample

The research involved 333 fourth-grade students at the age of 10 in Zagreb (the capital of Croatia), and in Zagreb County.

This age group is believed to be suitable for research into the effects of inquiry-based teaching for two reasons. Firstly, it is the age at which children exhibit natural curiosity which motivates them to examine, explore, create, and discover the world around them. It is therefore important for the teaching process in this stage of a child's development to meet their cognitive needs and spark their inquiring spirit by offering numerous incentives for exploring and discovering the unfamiliar, as well as different ways of reaching those discoveries. Armstrong [29] argues that there is a need to introduce developmentally appropriate practices for students aged 7–12 in schools, among which he emphasizes activities focused on identifying the features of the real world, use of authentic learning materials, and learning based on encounters with the real world, which will lead to the shaping of ideas, discovering, reflecting, and observing. Those practices are also the main characteristics of inquiry-based teaching and were applied in the course of this experiment. Another reason why this specific age group was selected derives from the fact that the curriculum for the science class in the 4th grade of primary school offers exceptional possibilities for realization of the suggested topics by means of inquiry-based teaching.

The sample of respondents is a non-probabilistic appropriate sample. The use of non-probability sampling methods makes it impossible to determine the likelihood that a person will be included in the sample or to determine whether a person has any probability of being included. Even though probabilistic samples have more advantages, non-probabilistic ones are nonetheless used in numerous research studies due to objective reasons. Their greatest disadvantage is the fact that the inability to determine the likelihood of a person being selected into the sample makes it impossible to define how representative the sample is, thus reducing the possibility of generalizing the conclusions which will be drawn from the research, which was a disadvantage in this particular research.

Students' parents were informed of the purpose of this research, after which they provided their written consent prior to their child's participation in the study. The study protocol was approved by the ethics committee of the school.

This paper is based on experimental research which is appropriate for evaluating the effects of inquiry-based learning, as it allows one to establish a connection between the actions employed in teaching and the corresponding results that students achieve on account of those actions.

Students were divided into two groups: the experimental group ($N$ = 164) and the control group ($N$ = 169). The groups were equal with respect to gender, year of birth, grade in science, and the general achievement score at the end of the third grade. The research encompassed eight urban and suburban schools. In each school, there was one experimental group and one control group. The respondents were selected on the basis of the random class selection criterion and students were randomly assigned to classes at the beginning of their education. The research was conducted in eight different schools divided into two larger groups (urban and suburban schools) to try to reduce the possible impact of the specific school atmosphere and experience with teaching science and to ensure a greater possibility of generalization of the data collected by means of this research. Before the experiment was conducted, the teachers were interviewed to examine the working conditions in the schools and previous teachers' experience with science class implementation. It was established that the schools had equal conditions for the implementation of inquiry-based teaching and that teachers' experiences with teaching science did not differ significantly among the eight schools. In all selected schools, students had very few opportunities to experiment on their own. Previous experiences with teaching science typically involved watching demonstration experiments being conducted by teachers and, very rarely, group work. Due to the fact that the examined schools did not differ significantly in terms of the organization of the science class and working conditions and that, so far, they mostly had not encouraged students to participate in activities which require independent designing and implementation of experiments as a part of inquiry-based learning, it can be assumed that the students' previous school experience did not significantly impact the result of this experiment.

Sixteen first- to fourth-grade teachers with similar qualification levels and length of service also participated in the experiment. The *Life Conditions* teaching unit was selected for the implementation of inquiry-based teaching in this science class experiment, as both groups of students planned an equal number of teaching hours for the said unit. Experimental research with parallel groups included groups of students who were exposed to inquiry-based teaching of primary science for three months (experimental group) and groups of students who were, at the same time, exposed to traditional teaching of the same curriculum content (control group). Teachers in the control groups taught the class in accordance with their usual class preparation and in compliance with the methodology and content instructions provided by the teaching curriculum. Teachers in the experimental groups taught the class in accordance with a specially designed preparation for a science class delivery, which was in line with the curriculum in terms of its content but the focus was placed on the use of elements of inquiry-based teaching and natural science methods: observation, description, comparing, data collection, data recording, data presentation, drawing conclusions and data interpretation, forming assumptions, planning autonomous research, experimenting, independent use of literature, and writing reports on the research.

Independent variables have been introduced in the experiment with parallel groups in order to determine a change in the education process, which is, in itself, a dependent changeable variable. Given the fact that each group was characterized by its own experimental factor (the control group was taught by means of traditional, lecture-based teaching, while inquiry-based teaching was implemented in the experimental group), inquiry-based teaching and lecture-based teaching were the independent variables in this experiment. Inquiry-based teaching as an independent variable included the use of the aforementioned natural science methods in teaching science.

The dependent variable in this research, which was tested with regard to the effect of independent variables, was the development of students' learning to learn competence. The learning to learn competence encompassed its constituent constructs: awareness of the importance of learning and different learning strategies, and the ability to use different learning strategies and overcome obstacles in learning, as well as attitudes towards independent learning and its importance for the life of each individual.

The research consisted of two stages. The first stage was the initial testing, in which all students completed a questionnaire about their learning habits and experiences, which allowed an estimation of

their learning to learn competence before further investigation and introduction of the experimental factor in their science classes. In the second stage, the experimental factor was introduced in teaching (inquiry-based learning in experimental groups; traditional teaching in control groups). The experiment lasted for three months, during which students had science lessons three times a week. To ensure the uniformity of the teaching style in inquiry-based and traditional teaching, each teacher ($N = 16$) received specially prepared lesson plans for each educational topic. During the final stage of the research, the same instrument was used as during the initial testing.

### 2.2. Instruments for Testing the Students' Learning to Learn Competence

The development of the students' learning to learn competence was examined by means of an adapted questionnaire which was constructed for fourth-grade students in 2007 by the Institute for Social Research in Zagreb, Center for the Research and Development of Education [6]. The questionnaire was adapted and amended for the purpose and needs of this research.

The instrument consisted of four parts which determined: (1) students' perception of and experience regarding the importance and usefulness of learning primary science for their present and future life (item example: *Everything I learn during science class, I will be able to apply in everyday life situations*); (2) students' motivation and interest to learn science (item example: *Learning science in primary school is very important to me; I am very much interested in scientific topics*); (3) how students perceive the learning of primary science in different circumstances and which learning circumstances they consider the most suitable (item example: *During a science class, I learn best when we perform group experiments*); (4) what strategies students use and how often they use them in order to increase the effectiveness of their own learning and what their learning habits are (item example: *When I learn science, I create a mental map with important information from the text*).

Each sub-scale consisted of 8 items. In the first 3 sub-scales, the respondents used a 4-point Likert scale (from 1—strongly disagree to 4—completely agree) to self-assess their learning experience in a science class, their motivation and interest to learn science, and the learning conditions they consider the most conducive to effective learning, such as working in groups, problem solving, or independent research. In the fourth sub-scale, the respondents used a 5-point scale (from 1—almost never to 5—almost always) to assess which learning strategies they used in order to improve their own learning efficiency and effectiveness (such as mental mapping, developing a learning plan, and linking with previously acquired knowledge) and what their learning habits were (such as searching for information on the Internet or skipping the unclear parts during learning). The same instrument was applied in the initial and final testing to check the effect of inquiry-based teaching of primary science on the development of the students' learning to learn competence. The internal consistency and reliability of the used scales (Cronbach's $\alpha$ coefficients), which was identified in the assessment of the aforementioned constructs of the learning to learn competence, was at a satisfactory or high level ($\alpha = 0.78$–$0.90$) for all the scales used in both situations (pre-testing and post-testing).

### 2.3. Data Processing Methodology

The connection between inquiry-based science teaching and the development of the students' learning to learn competence was determined by multivariate analysis of variance (MANOVA), where the dependent variable was the learning to learn competence, tested at two levels (initial and final testing), and the independent variable was the teaching strategy (inquiry-based teaching and traditional teaching) which was used in the primary science classes. The *t*-test was also conducted to determine the statistical significance of the difference in the results during the initial and final testing for each sub-scale, as well as the entire questionnaire.

## 3. Results

### 3.1. Initial Testing

As can be seen in Table 1, the arithmetic means of the answers of the control and experimental groups for different sub-scales, as well as the entire learning to learn questionnaire, in the situation of initial testing, are approximately equal.

**Table 1.** Comparison of the examination of experimental and control groups' learning to learn competence in the situation of initial testing.

|  | Group | N | M | SD | *t*-Test | df | *p* |
|---|---|---|---|---|---|---|---|
| Perception of learning | Control group | 169 | 14.78 | 3.57 | −0.33 | 331 | 0.74 |
|  | Experimental group | 164 | 14.90 | 2.68 |  |  |  |
| Motivation to learn | Control group | 169 | 15.40 | 3.78 | −0.42 | 331 | 0.68 |
|  | Experimental group | 164 | 15.57 | 3.63 |  |  |  |
| Learning strategies | Control group | 169 | 47.10 | 8.56 | −0.92 | 331 | 0.36 |
|  | Experimental group | 164 | 47.94 | 8.02 |  |  |  |
| Learning in different circumstances | Control group | 169 | 16.03 | 3.38 | 0.25 | 331 | 0.80 |
|  | Experimental group | 164 | 15.95 | 2.64 |  |  |  |
| Learning to learn competence (total) | Control group | 169 | 93.31 | 14.05 | −0.74 | 331 | 0.46 |
|  | Experimental group | 164 | 94.35 | 11.54 |  |  |  |

The results of the *t*-test show that there is no statistically significant difference in the learning to learn competence between the experimental and control group ($t = −0.74$; df $= 331$; $p = 0.46$), as well as in separate components of that competence, so it can be concluded that students in both groups had equal learning to learn competence before the implementation of experimental factors in teaching primary science.

### 3.2. Final Testing

As shown in Table 2, the arithmetic mean values of the answers provided by the control group and the experimental group for different sub-scales of the questionnaire, as well as the entire questionnaire, are different to a statistically significant degree in the final testing situation. The exception is the result of the "Motivation to learn" sub-scale, where the values of the arithmetic means of the control and experimental groups are approximately equal.

**Table 2.** Comparison of the examination of experimental and control groups' learning to learn competence in the situation of final testing.

|  | Group | N | M | SD | *t*-Test | df | *p* |
|---|---|---|---|---|---|---|---|
| Perception of learning | Control group | 169 | 15.37 | 3.43 | −5.68 | 331 | 0.00 |
|  | Experimental group | 164 | 17.24 | 2.47 |  |  |  |
| Motivation to learn | Control group | 169 | 14.93 | 3.59 | 1.08 | 331 | 0.28 |
|  | Experimental group | 164 | 14.55 | 2.89 |  |  |  |
| Learning strategies | Control group | 169 | 46.47 | 10.66 | −2.94 | 331 | 0.00 |
|  | Experimental group | 164 | 49.37 | 6.89 |  |  |  |
| Learning in different circumstances | Control group | 169 | 15.98 | 3.50 | −2.74 | 331 | 0.01 |
|  | Experimental group | 164 | 16.87 | 2.33 |  |  |  |
| Learning to learn competence (total) | Control group | 169 | 92.75 | 16.58 | −3.50 | 331 | 0.00 |
|  | Experimental group | 164 | 98.03 | 10.04 |  |  |  |

Table 2 shows that students in the experimental group had a higher arithmetic mean of answers on all sub-scales of the learning to learn questionnaire, except the "Motivation to learn" sub-scale. The *t*-test shows that there is a statistically significant difference with a risk of less than 1% in the learning to learn competence of students in the experimental and control group in the final test situation (*t* = −3.50; df = 331; *p* = 0.00), in favor of the experimental group. Based on these results, it can be concluded that students in the experimental group have a statistically significantly more positive experience of learning primary science, more often apply certain strategies that contribute to the quality of learning, and are more positive about learning in different circumstances compared with students in the control group, while their motivation to learn primary science in the situation of the final examination is equal.

*3.3. Comparison of the Difference in the Learning to Learn Competence between Students in the Control and Experimental Groups at Initial and Final Testing*

The results of multivariate analysis of variance (MANOVA) for the whole questionnaire show that students in the experimental group achieved a significantly higher level of scientific competence than the students in the control group (*F* = 7.03; df = 1; *p* < 0.05).

Furthermore, Table 3 and Figure 2 show that the statistical significance of the interaction of the two main effects was determined; i.e., it was confirmed that, in the final testing situation, the learning to learn competence of students from the experimental group was better to a statistically significant degree compared with students from the control group in the same situation (*F* = 6.31; df = 1; *p* < 0.05). From these results, it can be concluded that inquiry-based science teaching resulted in the development of better learning to learn competence in relation to traditional lecture-based teaching, so the hypothesis which claims that inquiry-based teaching of science in primary school would result in a rapid development of the students' learning to learn competence compared with traditional teaching is accepted.

**Table 3.** Statistical significance of major effects and interactions.

|  | *F* | **df** | *p* |
|---|---|---|---|
| Group | 7.03 | 1 | 0.01 |
| Competence | 3.44 | 1 | 0.07 |
| Competence*Group | 6.31 | 1 | 0.01 |

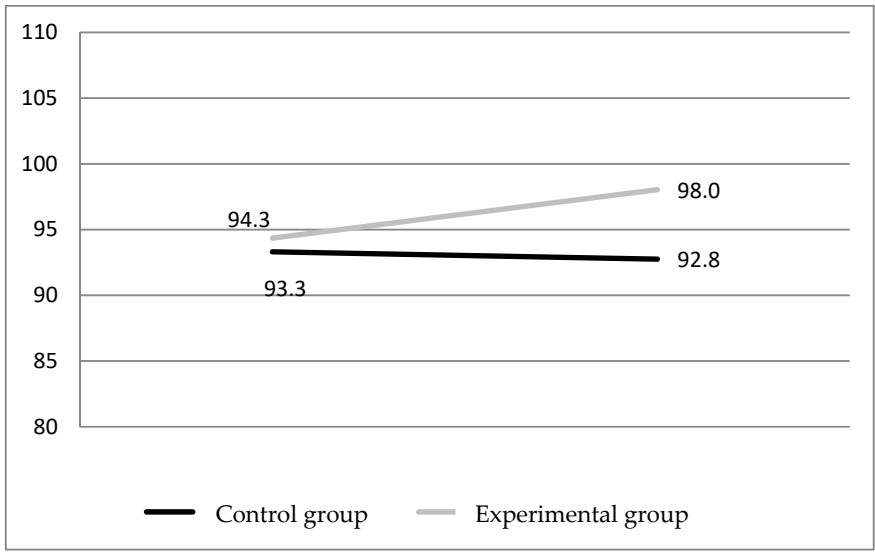

**Figure 2.** Interaction of group and learning to learn competence (total results).

## 4. Discussion

Whereas many studies investigate the effects of inquiry-based versus traditional lecture-based teaching on the students' academic achievements in primary science classes [30], few studies have considered its effect on the development of the students' learning to learn competence [4,31]. Such research is especially rare in the first educational stages. In addition, previous researches into the learning to learn competence have shown the possibility of its development during the educational process through the implementation of specific teaching strategies and active learning methods. Thus, this work focuses on how inquiry-based teaching, as opposed to traditional teaching, affects students' perception of learning, their motivation to learn, their process of learning in different circumstances, and the impact it has on the application of different learning strategies.

This research shows that inquiry-based teaching produced a better development of the students' learning to learn competence and that its effect on that development was significantly higher than that of traditional lecture-based teaching. Based on that finding, the hypothesis that inquiry-based teaching of primary science will result in an increased development of students' learning to learn competence (with regard to traditional teaching) has been proven. Better development of the students' learning to learn competence in the experimental group can be interpreted as a result of the students' active inclusion in planning the inquiry process and in thinking and reasoning about the learning objectives which they needed to achieve. It is important to emphasize that inquiry-based learning is a student-cantered approach, focusing on questioning, critical thinking, and problem solving. Learners are actively involved in formulating the question and posing a problem and make their own connections about what they are learning. This allows them to gain a deeper understanding than they would get by just memorizing and recalling facts and they are able to develop a passion for exploration and learning. Besides, the learning to learn competence implies that students in the process of learning begin from previous knowledge and life experience, which is the main postulate of constructivist learning incorporated in the basis of inquiry-based teaching. The positive impact of inquiry-based learning has mainly been determined in their perception of learning and their use of different learning strategies. These findings could provide valuable information for successful shaping of initial STEM (Science, technology, engineering, and mathematics) education, which often depends on the students' interest and motivation to learn.

The positive shift in the development of the learning to learn competence with the inquiry-based teaching shows its significant role in preparing students for lifelong learning. It can be assumed that a longer exposure of students to inquiry-based teaching would have an even more positive effect on the development of their learning to learn competence in primary science, because a short period of three months was enough to achieve a statistically significant positive shift in the development of this competence.

## 5. Conclusions

This research shows that inquiry-based teaching can contribute to the development of lifelong learning skills in 10-year-olds, which is extremely important in today's fast-changing world. That is why it is necessary to offer numerous opportunities for students to participate in research activities in their regular primary science classes, because the processes of independent or guided experimentation allow students to develop relevant learning skills and acquire new information. Participation in research activities offers a unique opportunity to simultaneously strengthen conceptual understanding of the area/topic of research, acquire research skills, learn new skills, and understand the process of learning; as such, it should be the essential activity in natural science education. In this research, inquiry-based learning led to an increased motivation to learn science, improved perception of the learning process, and more efficient use of learning strategies and handling of different learning conditions among students in the fourth grade of primary school.

When assessing the effects of inquiry-based teaching on the development of students' learning to learn competence, it is necessary to take into account that the development of the learning competence

has been analyzed by means of a survey in which students expressed their observations regarding the changes in their learning process in the science class. Subsequent research aimed at examining the development of this competence might further explore students' use of the learning to learn competence in specifically designed situations, and testing a larger sample. A longitudinal study would also provide an insight into the possibilities of developing this competence in science classes at the secondary stage of education.

In the Croatian educational system, which is currently undergoing the process of new curriculum implementation [32], the inquiry-based approach is an integral part and one of the main concepts of the interdisciplinary school subject called "Science and Social Studies" in the first four grades of primary school but the possibility of its implementation in other school subjects cannot be excluded. The present research proves that that inquiry-based teaching should be implemented in Science classes at the first educational stage (Grades 1 to 4 of primary school) because of its wide range of benefits for students. In that context, it is highly important to emphasize that the implementation of inquiry-based teaching requires appropriate teacher competencies that will allow its high-quality organization and realization through thoroughly planned research activities for students, allowing them to develop not only their natural science literacy but generic competences such as the learning initiative and independent learning, the ability to analyze and synthesize learning contents, the ability to plan and manage their time during learning, and information management skills. These are, for instance, motivation and readiness to apply inquiry-based teaching, knowledge about authentic scientific research, skills for its implementation, and positive beliefs about its application in the teaching process. It demands adequate teacher guidance through well-planned timed activities that allow the meaningful construction of concepts during a primary science class. Therefore, it is also necessary to ensure appropriate teacher training courses in this area, since previous research shows that teachers often do not have clearly defined learning strategies and have not been taught how to learn, thus demonstrating an inability to develop one of the most important academic competences [33,34], and that teachers' beliefs about learning affect how they implement the development of learning to learn competence in the classroom [35].

Based on these findings, it can be suggested that inquiry-based teaching should be applied as often as possible in primary science classes.

**Funding:** This research received no external funding.

**Acknowledgments:** I would like to thank the students, teachers, and principals of the schools in Zagreb and Zagreb County who were willing to participate in this research. Without them, this study would not have been possible.

**Conflicts of Interest:** The authors declare no conflict of interest.

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
