# Peer review of "Development of Students’ Learning to Learn Competence in Primary Science"

_education, doi:10.3390/educsci10110325_

Round 1

Reviewer 1 Report

The article deals with an interesting didactic problem. Although it has been researched many times in pedagogical practice and theory, distinctive elements with regard to the country's educational system would be interesting. The article does not adequately explain the structure of the sample as well as the structure of the instrument used in the research. Independent and dependent research variables have not been explained precisely enough. For example, we only know that the respondents are fourth grade elementary school students, with no precise insight into age; we know in which country and county the students attend school, but we do not know which groups are in rural and urban areas; we do not know whether the conditions in these groups are equal. We do not have data on the sampling method. The conclusion of the research is logical, but insufficiently nuanced considering the elements that are missing in the methodological approach.

Author Response

Thank you for the review.

All changes in the paper are red.

Point 1. The paper is supplemented by the explenation of the sample structure, as well as with the explenation of the instrument´s structure (highlighted as red text under the title 2.1. Research sample, and 2.2. Instrument for testing the students´learning to learn competence).

Point 2. The paper is supplemented by precisely explenation of independent and dependent variables(from lines 312 to 325)

Point 3. It is explained that students from urban and suburban schools participated in the research, and why schools in rural settings were not included in the sample (red highlighted text from lines 273 to 284)

Point 4. It is explained the age of the respondents, and why students of that age were chosen for this research (red highlighted text from lines 240 to 253)

Point 5. It is explained how we know that conditions in experimental and control group are equal (red highlighted text lines from 284 to 300)

Point 6. The samplig method is explained (red text - lines 254 to 263).

Point 7. The conclusion is supplemented with regard to added parts in the paper (red highlighted text in conclusion -462-504).

All changes in the paper are highlighted in red.

Reviewer 2 Report

The topic of the article has certain importance within the field of Primary Education as it involves a new concept of teaching Science. Nevertheless, the article needs to be improved:

  1. Deep revision of English. It seems it has been translated word-by-word froma different language.
  2. Use of academic writing. In academic writing the use of 1st person singular/plural (I/we) should be avoided.
  3. Punctuation marks should also be revised: commas, colons, semicolons, quotation marks, apostrophes, hyphens...
  4. Objectives should be better explained. In scientific articles, rigour is essential.
  5. Both dependent and independent variables should be better explained.
  6. Conclusions need to be improved as they don't give an answer to objectives. They don't show a clear relationship.
  7. References are not enough to support the study, they'd need to be expanded.
  8. The sample is small, taking into account the topic presented.

Author Response

Thank you for the review.

Point 1. Proofreading of the article was made. Parts of the paper which are corrected, and new parts of the paper, are highlighted in red.

Point 2. Parts of the paper in which was used 1st person plural (we) have been corrected.

Point 3. Punctation marks (commas, colons, semicolons, quotation marks, aphostrophes, hypens) are revised. All changes are highligted in red.

Point 4. Objectives are better explained (from lines 227 to 235, red highlighted text).

Point 5. The paper has been revised with precisely explenation of independent and dependent research variables (from lines 300 to 325, red text).

Point 6. The conclusion is revised in regard to added parts of the paper and show relationships with these parts (from line 462-504, red text).

Point 7. References are revised (New references: 2, 3, 15, 22, 23, 24, 25, 33, 34, 35).

Point 8. Explenation of the sample size: Experimental research with parallel groups is very complex and demanding research method, and requires good control of variables. Therefore, it was not possible to conduct research on a larger sample. In the context of Croatian education system, this sample size is satisfactory, taking into account the research method which was used. Because of the importance of this topic for the Croatian education system, in the future, we will try to conduct research on a larger sample.

Round 2

Reviewer 1 Report

Corrections of the text, in line with the remarks from the first review, have been made.

Author Response

English proofreading was made.

There were no other suggestions for changes from the first reviewer.

Reviewer 2 Report

Please see the attachment, especially the yellow highlight part.

Author Response

  1. Line 2 - New version of title: Development of Students´ Learning to Learn Competence in Primary Science
  2. Line 58 The suggestion for the changes in Table are done. „Learning to learn“ is placed as title
  3. Line 111 – comma is deleted
  4. Line 190 – „bad“ has been replaced with “inappropriate“
  5. Line 238 – Added: ...in Zagreb (capital of Croatia), and in.....
  6. Line 254-255 The sentence was changed to: The use of non-probability sampling methods makes it impossible to determine...
  7. Line 260. Revised grammar - „makes it impossible“
  8. Line 262 – The last sentence is deleted
  9. Line 267 – Revised: „as it allows to establish“
  10. Line 269 –„The“ is deleted.
  11. Line 274 – lines from 277-280 (from previous document) – unnecessary parts are deleted
  12. 278 (previous document 285) – Revised English: „Before the experiment was conducted, the teachers were interviewed to examine the working conditions in the schools and previous teachers’ experience with Science class implementation.“
  13. 286 – „the“ is deleted
  14. 287 – „for implementation“ is replaced with „for the implementation of“
  15. 304-305 – „delieverd the class“ is replaced with “taught“
  16. 313 – revised English „Independent variables have been introduced in the experiment with parallel groups in order to determine a change in the education process which is in itself a dependent changeable variable. Given the fact that each group...“
  17. 314 „seen“ is replaced with “given the fact that...“
  18. 326 article „ a“ instead of „the“ in front of „questionnaire“
  19. 331 – during which
  20. 334 Revised English „During the final stage of the research the same instrument was used as during the initial testing.“
  21. 389 – „as can be seen“ from is replaced with „ As shown in....“
  22. 392 – Revised English - Motivation to learn
  23. 414- Revised English – in graph caption „Control group“ is aligned with „Experimental group“
  24. 433 - "this paper investigated" is replaced with "this work focuses on...."
  25. 434 „the“ is erased
  26. 435 „How it impact“ is replaced with „and the impact it has in the application of...“
  27. 438 „our hypothesis“ is replaced with „the hypothesis“
  28. 454- Revised English „motivation to learn“
  29. 462 „Indicates“ is replaced with „shows“
  30. 466 – revised grammar – allow
  31. 475 –„in this research“ is deleted
  32. 476 – „was“ is replaced with „has been analysed“
  33. The new term is „longitudinal study“
  34. 486 – „indicates that“ is replaced with „proves that...“